# NAADP-Evoked Ca^2+^ Signaling Leads to Mutant Huntingtin Aggregation and Autophagy Impairment in Murine Astrocytes

**DOI:** 10.3390/ijms24065593

**Published:** 2023-03-15

**Authors:** Cássia Arruda de Souza Pereira, Natalia de Castro Medaglia, Rodrigo Portes Ureshino, Claudia Bincoletto, Manuela Antonioli, Gian Maria Fimia, Mauro Piacentini, Gustavo José da Silva Pereira, Adolfo Garcia Erustes, Soraya Soubhi Smaili

**Affiliations:** 1Departament of Pharmacology, Escola Paulista de Medicina, Universidade Federal de São Paulo, São Paulo 04044-020, Brazil; 2Instituto de Ciências Ambientais, Químicas e Farmacêuticas, Universidade Federal de São Paulo, Diadema 09913-030, Brazil; 3Department of Epidemiology, Preclinical Research and Advanced Diagnostics, National Institute for Infectious Diseases IRCCS “L. Spallanzani”, 00149 Rome, Italy; 4Department of Biology, University of Rome “Tor Vergata”, 00133 Rome, Italy; 5Department of Molecular Medicine, University of Rome “Sapienza”, 00185 Rome, Italy

**Keywords:** Huntington’s disease, lysosome, NAADP, two-pore channels

## Abstract

Huntington’s disease (HD) is a progressive neurodegenerative disease characterized by mutations in the huntingtin gene (mHtt), causing an unstable repeat of the CAG trinucleotide, leading to abnormal long repeats of polyglutamine (poly-Q) in the N-terminal region of the huntingtin, which form abnormal conformations and aggregates. Alterations in Ca^2+^ signaling are involved in HD models and the accumulation of mutated huntingtin interferes with Ca^2+^ homeostasis. Lysosomes are intracellular Ca^2+^ storages that participate in endocytic and lysosomal degradation processes, including autophagy. Nicotinic acid adenine dinucleotide phosphate (NAADP) is an intracellular second messenger that promotes Ca^2+^ release from the endo-lysosomal system via Two-Pore Channels (TPCs) activation. Herein, we show the impact of lysosomal Ca^2+^ signals on mHtt aggregation and autophagy blockade in murine astrocytes overexpressing mHtt-Q74. We observed that mHtt-Q74 overexpression causes an increase in NAADP-evoked Ca^2+^ signals and mHtt aggregation, which was inhibited in the presence of Ned-19, a TPC antagonist, or BAPTA-AM, a Ca^2+^ chelator. Additionally, TPC2 silencing revert the mHtt aggregation. Furthermore, mHtt has been shown co-localized with TPC2 which may contribute to its effects on lysosomal homeostasis. Moreover, NAADP-mediated autophagy was also blocked since its function is dependent on lysosomal functionality. Taken together, our data show that increased levels of cytosolic Ca^2+^ mediated by NAADP causes mHtt aggregation. Additionally, mHtt co-localizes with the lysosomes, where it possibly affects organelle functions and impairs autophagy.

## 1. Introduction

Huntington’s disease (HD) is a rare autosomal dominant neurodegenerative disease that affects 2.71 individuals per 100,000 [1,2]. HD is clinically characterized by a triad of symptoms that includes motor changes, progressive cognitive loss and psychiatric disorders. Choreic movements is the most characteristic symptom of the disease and consists of involuntary, random and rapid movements in the distal extremities of the limbs [3]. In most cases, HD affects patients aged between 30 to 50 years, and the neurodegeneration progresses irreversibly over a period of 15 to 20 years [4,5].

HD is caused by mutation in the gene encoding huntingtin (Htt), resulting in an expansion of the CAG trinucleotide on the short arm of chromosome 4p16.3 [6]. This mutation in the Htt gene leads to the abnormal expression of a long chain of polyglutamine (poly-Q) in the N-terminal portion [7], which encodes the mutant form of huntingtin (mHtt) [6]. CAG repeats are variable, while in the non-affected population it ranges from 9 to 35 repeats; in HD patients, these repetitions range from 36 to 55 (even more up to 100) [8].

The mechanisms that lead to neuronal death and the development and progression of HD are related to a toxic gain in mHtt function [9]. It is known that mHtt is prone to form protein aggregates, generate toxic fragments of the N-terminal region, interfere with gene transcription and compromise protein degradation systems [10]. Additionally, HD is also associated with deregulation of calcium (Ca^2+^) signaling and homeostasis [11,12,13,14]. Studies have shown that mHtt can interact with different Ca^2+^ channels, such as voltage-gated Ca^2+^ channels (VGCC) located in plasma membrane [15], and the type 1,4,5-triphosphate receptor 1 (IP_3_R1) in the endoplasmic reticulum membrane [12]. Additionally, mHtt can also interact with the outer mitochondrial membrane, affecting the mitochondrial Ca^2+^ buffering and leading to mitochondrial dysfunctions, which includes the opening of the mitochondrial permeability transition pore, causing mitochondrial depolarization and, eventually, the activation of apoptotic pathways of cell death [16]. In a previous study performed by our group, glutamate-induced Ca^2+^ release is higher in cortical slices, corpus callosum and striatum from transgenic R6/1 mice, an in vivo model of HD, when compared to slices from control animals [17].

Lysosomes are acidic organelles whose main function is the degradation of macromolecules, participating in endocytosis, secretory pathways and autophagy [18,19,20]. Additionally, lysosome and acidic organelles such as endosome play an important role in Ca^2^-storage and intracellular signaling [21,22,23]. The efflux of Ca^2+^ from lysosomes is mediated by ion channels present in the membrane, such as Two-Pore channels (TPCs), which are channels belonging to a superfamily of voltage-gated ion channels [22]. Evidence has revealed that TPCs are activated after their interaction with nicotinic acid adenine dinucleotide phosphate (NAADP) [24,25,26,27]. NAADP is a pyridine nucleotide and a potent Ca^2+^ mobilizer that was identified through studies using sea urchin egg homogenates [28,29]. Evidence suggests that the Ca^2+^ mobilized by NAADP comes from acid Ca^2+^ stores. This evidence is supported by the description of TPCs, as well as their location in acid stocks and their permeability to Ca^2+^, suggesting them as candidates for NAADP-interacting receptors, and, indeed, many studies have shown data consistent with this hypothesis [24,25,30]. Likewise, studies conducted by our group also contributed to describing NAADP as the main agonist of TPCs, where gastric smooth muscle cells and astrocytes were stimulated with NAADP, and the Ca^2+^ released was derived from lysosome and through the TPCs’ receptors [31,32,33]. In addition to its action on Ca^2+^ release, NAADP had a role in inducing autophagy. Pereira and colleagues (2011) demonstrated that NAADP can modulate autophagy in astrocytes through the activating TPC2 receptors since treatment with NAADP promoted the accumulation of autophagic markers LC3-II and Beclin-1; moreover, overexpression of a non-functional TPC2 construct impaired the effects of NAADP [31]. In another study, similar results were found in astrocytes after stimulation with glutamate, suggesting that the induction of autophagy mediated by glutamate involves the participation of NAADP/TPCs signaling [33].

In addition, several studies have correlated the involvement of NAADP with dysfunctional autophagy in neurodegenerative diseases. For example, it has been reported that the overexpression of the protein leucine-rich repeat kinase 2 (LRRK2) promotes the increase of autophagy through Ca^2+^-dependent protein kinase. Kinase-β (CaMKK-β)/adenosine monophosphate (AMP) activated the pathway of protein kinase (AMPK), which is Ca^2+^ dependent. These effects were similar to the addition of NAADP and were reversed with Ned-19, an NAADP antagonist [34], or through overexpression of a non-functional construct of TPC2 [35]. However, there is no published study that correlates NAADP and HD-mediated Ca^2+^ signaling. Importantly, autophagy is the only mechanism that cells have to remove protein aggregates, including mHtt; however, the impairment of autophagic function facilitates mHtt accumulation and aggregation, as well as its cytotoxic effects [36]. In this scenario, the NAADP-mediated autophagy could represent an alternative to rescuing defective autophagy, enhancing degradation of mHtt and reducing its cytosolic levels.

Here, we investigated the effects of NAADP/TPC signaling on autophagy and huntingtin protein aggregation in astrocytes. For this purpose, the effect of NAADP in cells overexpressing mHtt-Q74 and the link between TPCs receptors were studied. The effect on mHtt aggregation and autophagy blockade, which may decrease and affect neuroprotection, was also explored.

## 2. Results

### 2.1. Overexpression of mHtt-Q74 Enhances NAADP-Mediated Ca^2+^ Release

NAADP is a second messenger capable of interacting with TPCs receptors in acid organelles and promoting Ca^2+^ release [32]. Several studies have associated alterations in Ca^2+^ and NAADP signaling with neurodegenerative diseases; however, this correlation in HD is still unknown. Here, to investigate the connection of NAADP/TPCs and mHtt-Q74 signaling, a murine astrocytes cell line overexpressing pEGFP-mHtt-Q74 or pEGFP (empty vector as control) was loaded with Fura-2 AM. Cells were stimulated with the nicotinic acid adenine dinucleotide phosphate, acetoxymethyl ester (NAADP-AM, 100 nM) in the presence and absence of Ned-19 (1 µM), the antagonist of TPCs. Representative records show that NAADP stimulation promoted an increase in cytosolic Ca^2+^ in astrocytes overexpressing mHtt-Q74, when compared to the control group (Figure 1a). Quantification of maximum Fura-2 fluorescence demonstrates increases of cytosolic Ca^2+^ in cells overexpressing mHtt-Q74, whereas Ned-19 pre-treatment significantly reverses NAADP evoked-Ca^2+^ signals (Figure 1b).

### 2.2. NAADP Induces mHtt-Q74 Aggregation

Overexpression of mHtt-Q74 in astrocytes promoted changes in NAADP-mediated Ca^2+^ signaling. Thus, we investigated the possible effect of NAADP on mHtt-Q74 aggregation, as well as the participation of Ca^2+^ in this process. Astrocytes overexpressing pEGFP-mHtt-Q74 were treated with NAADP-AM (100 nM, agonist of TPCs), Ned-19 (1 µM, antagonist of TPCs) and BAPTA-AM (10 µM, cytosolic Ca^2+^ chelator) and analyzed under a confocal microscopy. The treatment of NAADP-AM induced the aggregation of mHtt-Q74, when compared to the untreated group (Figure 1c). A large amount of aggregates were distributed in the cytoplasm and in the perinuclear region. To assess the role of NAADP in mHtt-Q74 aggregation, astrocytes were pre-treated with Ned-19, followed by the addition of NAADP-AM. Representative images demonstrate that Ned-19 reversed the NAADP-mediated effect by inhibiting mHtt-Q74 aggregation compared to the NAADP-treated group (Figure 1c). Finally, the role of Ca^2+^ in the aggregation of mHtt-Q74 was also evaluated, using the Ca^2+^ chelator BAPTA-AM, pre-incubated for 30 min, followed by NAADP-AM stimulation. Interestingly, mHtt-Q74 aggregation was suppressed in the presence of BAPTA-AM, as no aggregate was observed in the analyzed cells (Figure 1c). The quantification of mHtt-Q74 aggregates is shown in Figure 1d, where astrocytes stimulated with NAADP-AM showed a significant increase in aggregates, when compared to the control group. Likewise, a statistical difference was detected when the NAADP-stimulated group was compared to groups pre-treated with Ned-19 and BAPTA-AM, which suppressed mHtt-Q74 aggregation.

Previous results from our group demonstrated that TPC2 is located mainly in lysosomes; in addition, NAADP-mediated Ca^2+^ release is greater in cells that overexpress the TPC2 receptor than in cells overexpressing the TPC1 receptor [30,31,32]. Thus, to evaluate the participation and role of the TPC2 channel in mHtt-Q74 aggregation, this channel was silenced in astrocytes overexpressing mHtt-Q74, using the siRNA-TPC2 (5′-3′ sequence: GGAAACCUCUUGUCUAUUUTT). Silenced TPC2 cells (Figure 1e) were treated with NAADP-AM (100 nM), and pEGFP-mHtt-Q74 aggregation was evaluated by confocal microscopy. As demonstrated in representative images (Figure 1f) and statistical analysis (Figure 1g), in the astrocytes where TPC2 was silenced (siTPC2), the NAADP-AM did not induce mHtt-Q74 aggregation. However, mHtt aggregation was observed in scramble cells.

### 2.3. mHtt-Q74 Colocalizes with TPC2 Receptor

It has been reported that mHtt overexpression and presence interact with Ca^2+^ channels, such as IP3R and RYR, modifying their function and cellular Ca^2+^ homeostasis [12,13]. Additionally, the data presented in Figure 1a,b demonstrate that mHtt-Q74 affects NAADP-mediated Ca^2+^ release via TPCs channels. In this way, to verify mHtt-Q74’s co-localization and interaction with TPC2 receptors, astrocytes overexpressing mHtt-Q74 were transfected with the RFP-TPC2 construct and observed in confocal microscopy (Figure 2a). Confocal images indicated that mHtt-Q74 partially colocalizes with TPC2 receptors, as demonstrated in the merged and zoomed images. To quantify the colocalized eGFP and RFP fluorescence pixels, the colocalization coefficient was applied, and the weighted colocalization coefficients were calculated. Data showed that mHtt-Q74 partially colocalizes with TPC2, when compared to the control group (RFP-empty).

### 2.4. mHtt-Q74 Inhibits Autophagy in Astrocytes

TPC2 has been described as the NAADP receptor in acidic compartments, such as the lysosomes [31,32]. Thus, the autophagic flux mediated by NAADP was evaluated in non-transfected astrocytes, in order to analyze the activation of autophagy in the absence of mHtt-Q74. To overexpress TPC2, astrocytes were transduced with the Myc-TPC2 construct or mCherry (empty vector) if used as the control group. Western blot analysis demonstrated increased Myc expression in the group transduced with the Myc-TPC2 construct, when compared to the control (mCherry) (Figure 3a). Astrocytes were treated with NAADP-AM (100 nM) for 1 and 2 h, in the presence or absence of E64d + Pep A (10 μg/mL) in the last hour of treatment. The results demonstrated that the overexpression of TPC2 promoted the accumulation of LC3-II, which was potentiated after 1 h treatment with NAADP-AM (Figure 3b,c). The groups treated in the presence of E64d + Pep A had a robust accumulation of LC3-II when compared to the control. The p62 analysis showed no statistical differences (Figure 3d). Once the role of TPC2 in the activation of the autophagic flux was demonstrated, the same experiment was performed in astrocytes overexpressing mHtt-Q74. Cells were treated for 1, 2 and 4 h with NAADP-AM (100 nM), in the presence or absence of E64d + Pep A, followed by evaluation of autophagic flux. As demonstrated in Figure 3e,f, NAADP-AM did not induce increases in the autophagic flux in astrocytes overexpressing mHtt-Q74, even in cells treated with E64d + Pep A, indicating an autophagy blockade in the degradative step of autophagy in this model. The p62 evaluation did not demonstrate statistical differences (Figure 3g).

## 3. Discussion

The Ca^2+^ ion is an important intracellular second messenger, which is involved in several physiological processes, and its homeostasis is essential to the maintenance of cell functions and viability [37,38]. The intracellular concentrations of Ca^2+^ are finely regulated through the action of pumps and exchangers located in membranes of the major Ca^2+^-store’s organelle and through the action of intracellular second messengers, such as NAADP [39]. Many studies have shown the involvement of acidic organelles and NAADP in neurodegenerative diseases, such as Parkinson’s and Alzheimer’s disease [40,41], but the involvement is still unknown for the Ca^2+^/NAADP signaling pathway in HD. Therefore, this study investigates the role of NAADP-mediated Ca^2+^ signaling in an in vitro model of HD and its effects on the aggregation of the mutated huntingtin protein Q74 (mHtt-Q74).

Our data demonstrated that astrocytes overexpressing mHtt-Q74 are more sensitive and presented an increase in cytosolic Ca^2+^ levels after stimulation with NAADP-AM (100 nM) (Figure 1a,b). A similar effect was observed by Pereira and collaborators, who demonstrated that NAADP (100 nM and 1 μM) induced the release and, consequently, the increase of intracellular Ca^2+^ in astrocytes in primary culture [31]. In addition, NAADP-AM promoted a significant increase in mHtt-Q74 aggregates when compared to the control group. However, the combination of NAADP-AM with Ned-19 or BAPTA-AM leads to a reduction of mHtt-Q74 aggregates in astrocytes when compared to the group treated with NAADP-AM. A similar effect was observed when TPC2 was silenced in astrocytes; the formation of mHtt aggregates was reduced (Figure 1f–h). Together, these data suggest the participation of Ca^2+^ in mHtt-Q74 aggregation, which was reverted in the presence of calcium chelator (BAPTA-AM), as well as the possible participation of TPC receptors.

Deregulation in Ca^2+^ signaling is a common characteristic in neurodegenerative disorders, such as Parkinson’s disease and Alzheimer’s disease [42,43]. In Huntington’s disease, mHtt seems to interact with different types of Ca^2+^ receptors and channels, altering their functioning and modifying the intracellular Ca^2+^ signaling [12,13,44,45]. Our results demonstrated colocalization between mHtt-Q74 (pEGFP-mHtt Q74) and TPC2 channels in astrocytes overexpressing RFP-TPC2 constructs. These data suggest a possible physical interaction of mHtt with this receptor, affecting its functioning (Figure 2a,b). Similar results were obtained using the fibroblasts of a LRRK2 G2019S Parkinson’s patient, in which lysosomal morphology defects were corrected with molecular TPC2-silencing or pharmacological inhibition of TPCs [40].

It is well known that the interaction of NAADP with the TPCs can induce autophagy in astrocytes [31,33]. In this way, the autophagic flux mediated by NAADP was investigated in an in vitro model of HD. Initially, autophagic flux was evaluated in astrocytes overexpressing the TPC2 channel (Myc-TPC2) to characterize the NAADP-AM response. As expected, cells overexpressing TPC2 channels had an increase in the LC3-II autophagic marker (Figure 3b,c). However, in astrocytes overexpressing mHtt-Q74 and Myc-TPC2, no significant changes were detected in the levels of LC3-II and p62 proteins after the treatment with NAADP-AM at different time-points, indicating the reduction of autophagy in our model (Figure 3e,f). These data may indicate that mHtt affects autophagic signaling and machinery, blocking autophagy in these cells, suggesting that autophagy-mediated NAADP signaling is affected by mHtt. In fact, the reduction of autophagic activity is a common feature in neurodegenerative diseases, where there is an accumulation of protein aggregates [46,47].

The correlation between autophagy and HD has been widely studied, as autophagy induction is often proposed as a therapeutic target of HD in many studies [4,48,49,50]. The evaluation of autophagy markers performed in patients with HD, the expression levels of genes related to autophagy are increased, such as LAMP2A, LC3I and ULK2 [49]. Studies have shown that wild type Htt plays an important role in autophagy, and the occurrence of mHtt interferes with normal huntingtin functions. This aspect was reported by Wong and Holzbaur, where wild type Htt could act as a regulator of autophagosome transport in neurons, and the depletion of wild type Htt or mHtt expression would lead to dysregulation of this transport and, consequently, would affect and interfere with autophagy degradation [51].

Taken together, our data demonstrated that mHtt-Q74 colocalizes with the TPC2 receptor in lysosomes, consequently affecting the physiological function of this channel, and induces the release of Ca^2+^ from this organelle. Increased levels of Ca^2+^ originating from the lysosome favor mHtt-Q74 aggregation. Additionally, data have demonstrated that there is inhibition of autophagy in astrocytes overexpressing mHtt-Q74. These results are compatible with the hypothesis that lysosomal homeostasis is important to inhibit the mHtt aggregation and to consider autophagy as a potential neuroprotective system.

## 4. Materials and Methods

### 4.1. Cell Culture, Transfections and Retroviral Infection

Immortalized astrocytes were cultured in DMEM (Dulbecco’s modified Eagle’s medium) with high glucose content, supplemented with 10% fetal bovine serum and 1% penicillin/streptomycin and kept at 37 °C in 5% CO_2_ humidified atmosphere. Briefly, primary astrocytes were obtained from 4-day-old Wistar rats and immortalized through overexpression of the T antigen [52].

To overexpress mHtt-Q74, astrocytes were transfected with the plasmid pEGFP-mHtt Q74 (3 μg, Cat #40262, Addgene, Watertown, MA, USA) using Lipofectamine 3000 (6 μL), according to the manufacturer’s protocol (Thermo Fisher Scientific, Waltham, MA, USA). The control group was prepared with astrocytes transfected with the plasmid pEGFP-empty (Cat #165830; Addgene, Watertown, MA, USA). As previously demonstrated in other studies, the overexpression of nonpathogenic huntingtin (Htt-Q23) does not promote protein aggregation or affect cellular viability, when compared to the mHtt-Q74 [53,54]. Therefore, in the present study, astrocytes transfected with the pEGFP-empty vector were used as the control group. Similarly, astrocytes were also transfected simultaneously with the pEGFP-mHtt Q74 (1.5 μg) and with the plasmid RFP-TPC2 (1.5 μg) using Lipofectamine 2000 (5 μL) according to the manufacturer’s protocol (Thermo Fisher Scientific). pEGFP-Q74 was a gift from David Rubinsztein (Addgene plasmid #40262; http://n2t.net/addgene:40262 accessed on 31 December 2022; RRID:Addgene_40262) and pEGFP was a gift from Koen Venken (Addgene plasmid #165830; http://n2t.net/addgene:165830 accessed on 31 December 2022; RRID:Addgene_165830).

The myc-tagged, full-length, wild-type TPC1 and TPC2 constructs were kindly provided by Sandip Patel (UCL, London, UK) [24] and cloned into a modified version of the pLPCX vector (Clontech, Mountain View, CA, USA) [55]. For virus production containing pLPCX-Myc-TPC2, 15 μg retroviral vector was co-transfected with 5 μg expression plasmid for the vesicular stomatitis virus G protein into the 293 gp/bsr cell line using the calcium phosphate method. After 48 h, the supernatant containing the retroviral particles was recovered and supplemented with 4 μg/mL polybrene and stored at −80 °C.

In addition to overexpression assay, TPC2 channel was also silenced in astrocytes overexpressing the mHtt. For this purpose, 1.0 × 10^5^ cells/well were co-transfected using Lipofectamine RNAiMAX (Life Technologies, Carlsbad, CA, USA) (7.5 μL) with pEGFP-mHtt Q74 (1.5 μg) and RNAi siTPC2 (20 nmol; Ambion, Life Technologies), corresponding to TPC2 target sequence (sequence 5′-3′: GGAAACCUCUUGUCUAUUUTT). A scramble sequence was used as a control.

### 4.2. Ca^2+^ Measurements

To analyze the cytosolic Ca^2+^ mobilization, astrocytes were cultured in 25-mm glass coverslips and loaded with the fluorescent ratiometric indicator Fura-2 AM (5 μM, Thermo Fisher Scientific Cat# F1201) and Pluronic F-127 (20%, Sigma-Aldrich, San Luis, MO, USA) for 30 min at 37 °C in fluorescence buffer (130 mM NaCl; 5.36 mM KCl; 1 mM MgSO_4_; 1 mM Na_2_HPO_4_; 1.5 mM CaCl_2_; 2.5 mM NaHCO_3_; 1.5 mM albumin; 25 mM glucose and 20 mM HEPES pH 7.4). Fura-2 fluorescence records were performed using a real-time fluorescence microscope (Nikon TE300, Nikon, Tokyo, Japan) coupled to a Coolsnap high-resolution digital camera (RoperSci, Princeton Instruments, Trenton, NJ, USA) at 40× magnification. Fura-2 fluorescence was recorded for 30 min (300 images), with a 6 s interval between images acquisition. The basal Fura-2 fluorescence was recorded in the first 2 min (20 images) of each experiment. Following the establishment of the baseline, cells were stimulated with NAADP-AM (100 nM), in the presence and absence of NED-19 (1 μM), that act as its antagonist. The acquisition of images, data extraction and analysis of regions of interest (ROIs) were performed using the Imaging software BioIP. Only the cells transfected with the pEGFP-mHtt Q74 were analyzed. The data are presented as the maximum fluorescence ratio of Fura-2, normalized according to basal fluorescence. Cells loaded with Fura-2 were excited at 340 nm and 380 nm, and the fluorescence emission was detected at 510 nm.

### 4.3. Western Blotting

For protein extraction, cells were lysed in RIPA buffer (150 mM NaCl; 1% NP-40; 0.5% deoxycholic acid; 0.1% SDS; 50 mM Tris pH 8.0; 2 mM MgCl_2_), supplemented with protease and phosphatase inhibitor cocktails, followed by centrifugation to remove cellular debris. Total proteins were quantified using the Bradford assay, and samples were prepared in NuPage LDS sample buffer (Thermo Fisher Scientific). A total of 15–40 μg of each protein sample was subjected to SDS-polyacrylamide gel electrophoresis and transferred onto a nitrocellulose membrane. Immunoblots were blocked in 5% nonfat dry milk for 30 min at room temperature, followed by overnight incubation with primary antibodies: anti-GFP (1:200; Santa Cruz Biotechnology, Dallas, TX, USA; Cat# sc-8334, RRID:AB_641123), anti-vinculin (1:1000; Cell Signaling Technology, Danvers, MA, USA; Cat# 13901, RRID:AB_2728768), anti-LC3B (1:2000; Cell Signaling Technology Cat# 2775, RRID:AB_915950), anti-SQSTM1/p62 (1:2000; MBL International, Woburn, MA, USA; Cat# PM045, RRID:AB_1279301). For internal control, anti-α-tubulin (1:10,000; Sigma-Aldrich Cat# T9026, RRID:AB_477593) and anti-GAPDH (1:10,000; Sigma-Aldrich Cat# G8795, RRID:AB_1078991) were used. Secondary antibodies were incubated 1 h at room temperature: anti-rabbit (1:5000; Thermo Fisher Scientific Cat# 31460, RRID:AB_228341) and anti-mouse (1:5000; Thermo Fisher Scientific Cat# G-21040, RRID:AB_2536527), conjugated with horseradish peroxidase (HRP). The membranes were developed with chemiluminescent substrate (ECL, PerkinElmer, Waltham, MA, USA), and the luminescence bands were captured in a high-resolution photodocumentation system (UVITEC Alliance 4.7, Cambridge, UK). The protein bands were analyzed by densitometry through evaluation of optical intensity using the Alliance software and normalized according each internal control band. Data are presented as the relative expression, comparing to control group.

### 4.4. mHtt Q74 Aggregation Assay

To quantify the number of aggregates per cell, the astrocytes were plated on 25-mm glass coverslips, and after transfection and/or treatment, the slides were observed. Images were acquired using an LSM 780—Axiovert 200 M confocal microscope (Carl Zeiss, Inc., Oberkochen, Germany) with a 63× oil immersion objective. For pEGFP-mHtt-Q74 visualization, excitation and emission, filters of 488/505–550 nm were used. ZenBlue Lite software (version 2.6) was used for data extraction. Approximately 10 images were acquired in each slide, and the fields were randomly chosen. These images were analyzed in relation to the number of aggregates that each cell presented.

### 4.5. Colocalization of mHtt-Q74 and TPC2 Receptor

To evaluate the colocalization of pEGFP-mHtt Q74 with RFP-TPC2 receptors, astrocytes were cultured in 25-mm glass coverslips and transfected with the mentioned plasmids. Live cells were observed with a confocal microscope (Zeiss LSM780 Axiovert 200 M, Carl Zeiss) in a 63× magnification. Images were acquired randomly from different locations, and the colocalization evaluation was performed in each cell separately. The analysis was performed using the software ZenBlue Lite software (ZEN Digital Imaging for Light Microscopy, RRID:SCR_013672, version 2.6, Carl Zeiss, Inc., Oberkochen, Germany), and the colocalization was calculated according to weighted colocalization coefficients. Excitation/emission: pEGFP-mHtt-Q74 (488/505 nm) and RFP-TPC2 (543/615 nm).

### 4.6. Statistical Analysis

Data are presented as mean ± standard error of the mean (SEM). All experiments were performed with a minimum of three independent replicates. Graphical representation, data and analysis were performed using GraphPad Prism (GraphPad Prism, Boston, MA, USA; RRID:SCR_002798), version 5.1, using one-way ANOVA followed by Tukey’s post hoc or Bonferroni post hoc or Student’s *t* test analysis. Differences between analyzed groups were considered significant when *p* < 0.05.

## Figures and Tables

**Figure 1 ijms-24-05593-f001:**
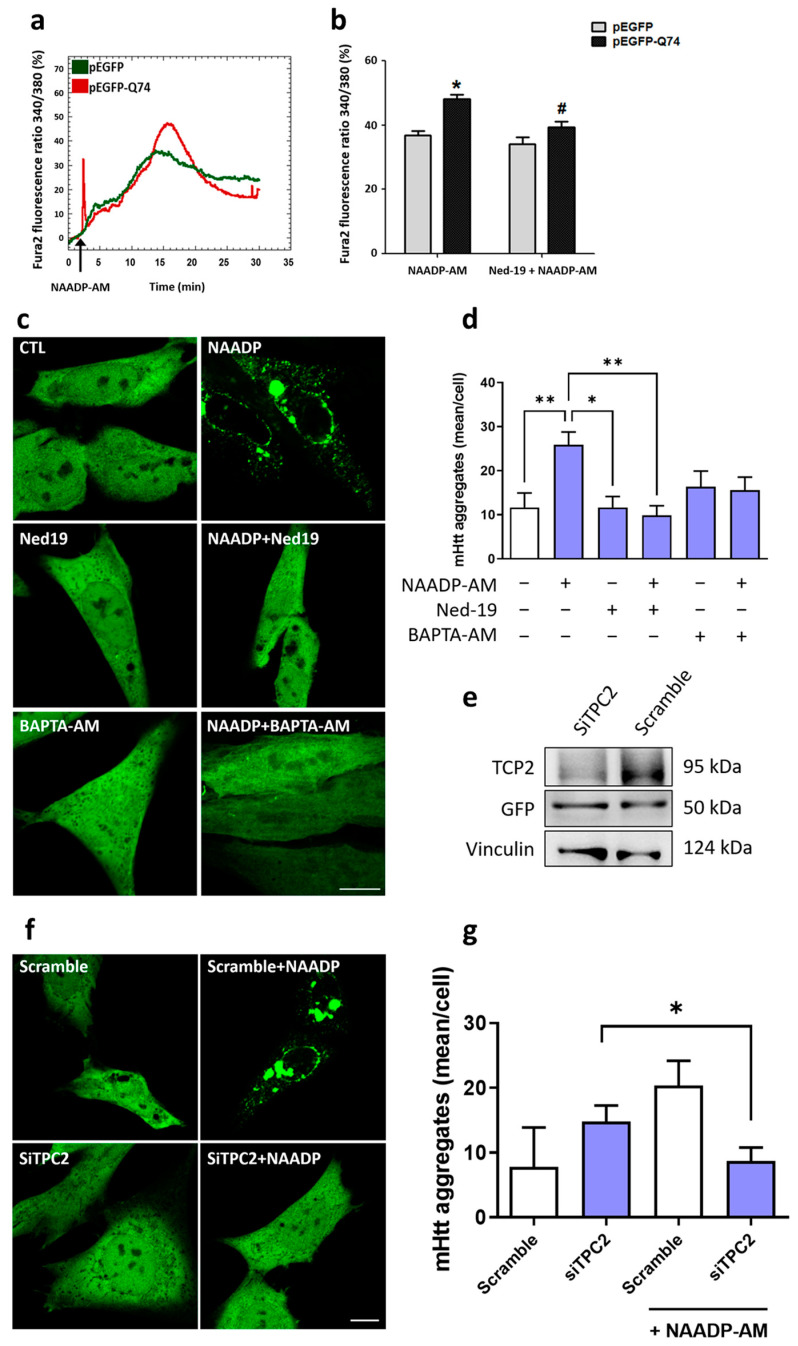
NAADP mediates Ca^2+^ mobilization and mHtt-Q74 aggregation in astrocytes. Astrocytes were transiently transfected with the pEGFP-mHtt-Q74 vector or pEGFP-empty vector, followed by evaluation of Ca^2+^ mobilization and mHtt-Q74 aggregation. Additionally, mHtt-Q74 aggregation was analyzed in astrocytes overexpressing or silenced for the TPC2 channel. Astrocytes overexpressing pEGFP (empty vector) or pEGFP-mHtt-Q74 were loaded with Fura-2 AM (5 μM), followed by stimulation with nicotinic acid adenine dinucleotide phosphate, acetoxymethyl ester (NAADP-AM, 100 nM) in the presence or absence of Ned-19 (1 μM), a TPCs antagonist. Fura-2 fluorescence was recorded for 30 min, considering the first minute of each experiment as basal fluorescence. (**a**) Representative records of the Fura-2 fluorescence ratio (340/380) of pEGFP (green line) and pEGFP-mHtt-Q74- (red line) overexpressing astrocytes. (**b**) Quantification of the maximum Fura-2 fluorescence ratio after the stimulation with NAADP-AM (100 nM), in the presence or absence of the Ned-19 (1 μM), normalized according to the basal fluorescence. *n* = 3. Data presented as mean ± SEM. One-way ANOVA, followed by Bonferroni post hoc. * *p* < 0.05; # *p* < 0.05. (**c**) Confocal microscopy of astrocytes overexpressing the pEGFP-mHtt-Q74 treated with NAADP-AM (100 nM) and Ned-19 (1 µM), both in the presence or absence of BAPTA-AM (10 µM), used as cytosolic Ca^2+^ chelator. Images were obtained using a Carl Zeiss LSM 780 confocal microscope (Carl Zeiss, Oberkochen, Germany), with 63× magnification. Scale bar: 10 μm. (**d**) Quantification of mHtt-Q74 aggregates (mean/cell) in the studied groups. *n* = 3. Data presented as mean ± SEM. One-way ANOVA, followed by Tukey’s post hoc. * *p* < 0.05; ** *p* < 0.01. To evaluate the participation of the TPC2 receptor in mHtt-Q74 aggregation, TPC2 was overexpressed or silenced in the astrocytes overexpressing pEGFP-mHtt-Q74. (**e**) Representative autoradiograms from western blotting analysis of astrocytes overexpressing pEGFP-mHtt-Q74 and silenced for TPC2 receptor, after incubation with anti-TPC2, anti-GFP or anti-vinculin antibodies. (**f**) Confocal images of astrocytes overexpressing pEGFP-mHtt-Q74 treated with NAADP-AM (100 nM), in control and silenced cells for the TPC2 receptor and (**g**) quantification of mHtt-Q74 aggregates in the evaluated groups. *n* = 2. Data presented as mean ± SEM. One-way ANOVA, followed by Tukey’s post hoc. * *p* < 0.05. Images were acquired using a Carl Zeiss LSM 780 confocal microscope, with 63× magnification. Scale bar: 10 μm.

**Figure 2 ijms-24-05593-f002:**
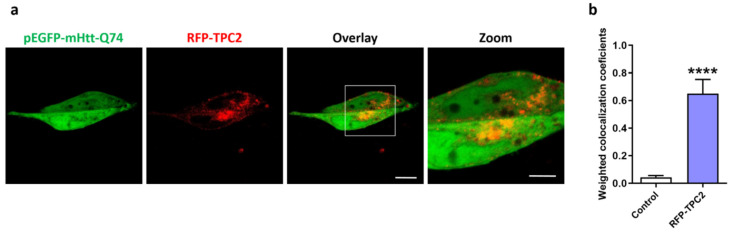
Evaluation of colocalization between the TPC2 receptor and mHtt-Q74. Astrocytes were transfected with pEGFP-mHtt-Q74 and RFP-TPC2 vectors, followed by evaluation of the cells in a confocal microscope. (**a**) Representative images of astrocytes overexpressing mHtt-Q74 protein (pEGFP-mHtt-Q74, green fluorescence) and TPC2 (RFP-TPC2, red fluorescence). The merged images show both fluorescence signals, and digital zoom (2×) on specific areas of interest was applied to demonstrate colocalization between the fluorescence signals. Images were obtained using a Carl Zeiss LSM 780 confocal microscope, with 63× magnification. Scale bar: 10 μm and 5 μm. (**b**) Quantification of colocalization between pEGFP-mHtt-Q74 (green fluorescence) and RFP-TPC2 (red fluorescence), calculated according to the weighted colocalization coefficient. *n* = 3. Data presented as mean ± SEM. Unpaired Student’s *t* test, **** *p* < 0.0001.

**Figure 3 ijms-24-05593-f003:**
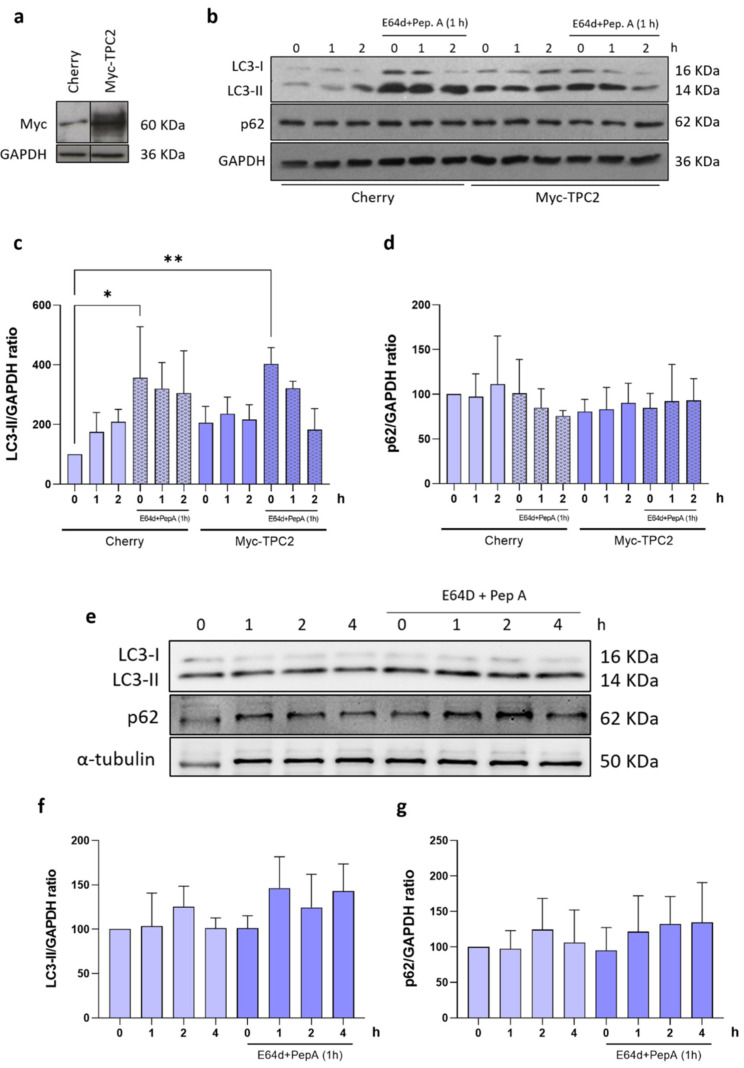
Evaluation of NAADP-induced autophagy in astrocytes overexpressing TPC2 or mHtt-Q74. Western blot analysis of the autophagic flux of astrocytes overexpressing Myc-TPC2 or pEGFP-mHtt-Q74 after treatment with NAADP-AM (100 nM), in the presence or absence of E64d/Pep A. Detection of autophagy was performed by analyzing the levels of LC3 and p62 levels after 1, 2 and 4 h of treatment with NAADP. (**a**) Representative autoradiograms of western blotting analysis of anti-myc and anti-GAPDH antibodies, demonstrating Myc tag protein overexpression in astrocytes transfected with the Myc-TPC2 construct. (**b**) Representative autoradiograms of western blotting analysis of anti-LC3, anti-p62 and anti-GAPDH antibodies in the astrocytes overexpressing Myc-TPC2. (**c**) Quantification of LC3-II and (**d**) p62 after treatments with NAAPD-AM in astrocytes overexpressing Myc-TPC2, both in the presence or absence of E64d/pepstatin A. The experiments were normalized in relation to the GAPDH internal control. *n* = 3. Data presented as mean ± SEM, One-way ANOVA followed by Tukey’s post hoc. * *p* < 0.05; ** *p* < 0.01. The autophagic flux was also evaluated in astrocytes overexpressing mHtt-Q74 after treatments with NAADP for 1, 2 and 4 h, in the presence or absence of E64d/pepstatin A. (**e**) Representative autoradiograms of the western blotting analysis of anti-LC3, anti-p62 and anti-α-tubulin antibodies. (**f**) Quantification of LC3-II and p62 (**g**) after treatments with NAAPD-AM in astrocytes overexpressing mHtt-Q74. The experiments were normalized in relation to the α-tubulin internal control. *n* = 3. Data presented as mean ± SEM, One-way ANOVA followed by Tukey’s post hoc.

## Data Availability

The data that support the findings of this study are available from the corresponding author upon reasonable request.

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
