# Peer review of "NAADP-Evoked Ca2+ Signaling Leads to Mutant Huntingtin Aggregation and Autophagy Impairment in Murine Astrocytes"

_ijms, 2023, doi:10.3390/ijms24065593_

Round 1

Reviewer 1 Report

In the article entitled: “ NAADP-evoked Ca2+ signaling leads to mutant huntingtin aggregation and autophagy impairment in murine astrocytes”, the authors study the effects of adenine dinucleotide phosphate (NAADP) and Two-Pore Channels (TPCs)   signaling on huntingtin protein aggregation in astrocytes and autophagy. For this purpose, on cells  overexpressing mHtt-Q74, the effect of NAADP on TPCs activity was studied.

They show that increased levels of cytosolic Ca2+ mediated by NAADP causes mHtt aggregation. Also, on the basis of the results, they state that mHtt co-localizes with the lysosomes, and this could affect organelle functions and impairs autophagy.

The results showed in Fig. 1 clearly show that overexpression of mHtt-Q74 stimulates NAADP-mediated Ca2+ release, as well as that NAADP induces mHtt-Q74 aggregation.

On the other hand, the results showed in Fig. 2 don’t clearly demonstrate  that   mHtt-Q74 colocalizes with TPC2 receptor: pEGFP-mHtt-Q74 staining is diffused all over cell cytoplasm, and more importantly, no lysosomes staining is present, so the assertion « our data demonstrated that mHtt-Q74 colocalizes with TPC2 receptor in lysosomes » is not supported by the results.

In order to make this work suitable for publication, it’s mandatory to improve images of fig.2 demonstrating colocalisation AND to add lysosome staining showing mHtt-Q74 lysosomial localization.

Here there are some minor corrections :

Line 23 : « add nicotinic acid  adenine dinucleotide phosphate » before the abbreviation

Line  52-62: Mitochondrial dysfunction is   involved in neurodegeneration that occurs in Huntington's disease, and these organelles are well known to be involved in Ca homeostasis. This point should be added.

Line 73 : which additional evidences ? specify and add bibliography

Line 82 : cancel « treatment »

Author Response

Comments and Suggestions for Authors

In the article entitled: “ NAADP-evoked Ca2+ signaling leads to mutant huntingtin aggregation and autophagy impairment in murine astrocytes”, the authors study the effects of adenine dinucleotide phosphate (NAADP) and Two-Pore Channels (TPCs)   signaling on huntingtin protein aggregation in astrocytes and autophagy. For this purpose, on cells  overexpressing mHtt-Q74, the effect of NAADP on TPCs activity was studied.

They show that increased levels of cytosolic Ca2+ mediated by NAADP causes mHtt aggregation. Also, on the basis of the results, they state that mHtt co-localizes with the lysosomes, and this could affect organelle functions and impairs autophagy.

The results showed in Fig. 1 clearly show that overexpression of mHtt-Q74 stimulates NAADP-mediated Ca2+ release, as well as that NAADP induces mHtt-Q74 aggregation.

We appreciate the comments made by the reviewer about our experiments and the data presented in our manuscript. We have addressed the questions about data and included the suggestions that we believe have improved the new version of the manuscript. We have changed and altered several points of our manuscript as presented below.

On the other hand, the results showed in Fig. 2 don’t clearly demonstrate  that   mHtt-Q74 colocalizes with TPC2 receptor: pEGFP-mHtt-Q74 staining is diffused all over cell cytoplasm, and more importantly, no lysosomes staining is present, so the assertion « our data demonstrated that mHtt-Q74 colocalizes with TPC2 receptor in lysosomes » is not supported by the results.

In order to make this work suitable for publication, it’s mandatory to improve images of fig.2 demonstrating colocalisation AND to add lysosome staining showing mHtt-Q74 lysosomal localization.

We agree with the reviewer, that pGFP-mHtt-Q74 staining is diffuse over cell cytoplasm. However, to perform the same experiment after mHtt aggregation could hide the objective and the main question: “how mHtt promotes the elevation of cytosolic calcium levels?”. In this way, we suggest that the adequate method to answer this question is evaluating the pEGFP-mHtt-Q74 signal when it is diffuse in cytoplasm. We understand that confocal microscopy has its limitations, however, the experiment performed to detect the colocaliaztion of pEGFP and RFP signal was conducted to minimize these limitations. We used living cells overexpressing specific markers, to provide a better acquisition of images and avoid photobleaching. Images were acquired in a 63x oil immersion objective followed by the capture using a 2x-digital zoom. The data analyses were performed using the colocalization coefficient, which is sensitive to detect the colocalization between two distinct fluorescence signals in the same pixel. These approaches to acquire and analyze the data were taken to minimize the limitations of confocal microscopy.

In a study published by our research group, Pereira et al. (2011) demonstrated that RFP-TPC2 signal is completed colocalized with lysosomes, which were stained with LysoTracker green, in primary astrocyte cultures. In the same study the authors also shown that TPC2 can modulate the release of calcium from lysosomes. The location of TPC2 in lysosomes and its role in calcium release is described by many authors, like Calcraft et al., (2009); Nature, doi: 10.1038/nature08030; Brailoiu et al., (2009), J Cell Biol., doi: 10.1083/jcb.200904073; Castro et al., 2022, Cells, doi: 10.3390/cells11182807). Therefore, the signal of RFP-TPC2 is an indication that lysosomes colocalize with mHtt-Q74 and RFP-TPC2 signal.

As pointed out, we have also included the minor corrections :

Line 23 : « add nicotinic acid  adenine dinucleotide phosphate » before the abbreviation.

The description was added in line 26.

Line  52-62: Mitochondrial dysfunction is involved in neurodegeneration that occurs in Huntington's disease, and these organelles are well known to be involved in Ca homeostasis. This point should be added.

We agreed with the reviewer, mitochondrial dysfunctions play a relevant role in Huntington’s disease neurodegenerative process. Thus, we have included a sentence in the line 65-69, that mention the mitochondrial relevance to HD, as well as its role in calcium homeostasis and cell death.

Line 73 : which additional evidences ? specify and add bibliography

The « additional evidence » mentioned in line 73 is cited in the next sentence, where we added additional informations to support that calcium is mobilized from acidic stores throught TPCs. The sentence was corrected in order to improve the information and the undestanding.

Line 82 : cancel « treatment »

The word « treatment » was removed from manuscript.

Author Response

This is very timely and interesting MS presenting a new data on HD (Huntington disease). It presents a new view on HD, the mostly enigmatic neurodegenerative disease (ND). Though the number of HD cases is rather low and depends on a certain geographical population, the course of this ND is not easier than in other NDs like Parkinson’s or Alzheimer’s diseases. Naturally, to cope with HD researches are concentrating on constructing animal or cell models enabling an express search and testing of putative therapeutics. In this respect the MS is rather instructing and providing a home-taken message. If so, a person in this field (as me), would not be concentrated on finding drawbacks of an MS, rather on improving its style. Therefore, I suggest minor corrections in sentences which in their present form lead away from understanding the causes and consequences.

We thank the reviewer and the comments about our manuscript. We have considered all the suggestions and corrected the mentioned points, and we wish to submit an improved version of our manuscript.

As pointed out, we have also included the minor corrections :

261-262: “Together, these data suggest the participation of Ca2+ in mHtt-Q74 aggregation of, which was reverted…”  The question- aggregation of what?,

We thank the reviewer pointing this typing error. The sentence was corrected (line 296-298), in order to demonstrate that in the presence of Ca2+, mHtt-Q74 shows aggregation which is reverted by calcium chelation.

275: “In this way, NAADP-mediated autophagic flux by was investigated” The question - flux by – what?

 The sentence was corrected in order to indicate that autophagic flux was mediated by NAADP (line 310-311).

297: “Taken together, our data demonstrated that mHtt-Q74 colocalizes with TPC2 receptor in lysosomes, and consequently might be driven Ca2+ release from this organelle”. Please, check, otherwise the sense is being lost.

We thank the reviewer to have pointed the lost sentence. We have checked the information in order to correct and clarify the take-home message of our work (line 333-335).

Reviewer 3 Report

In their paper entitled “NAADP-evoked Ca2+ signaling leads to mutant huntingtin aggregation and autophagy impairment in murine astrocytes”, the Authors report that an increase of cytosolic Ca2+ (mediated by the second messenger NAADP, that possibly directly interacts with two pore calcium channels, TPCs) causes mutated huntingtin (mHtt) to aggregate. Interestingly, they also show that mHtt co-localizes with the lysosomes, probably affecting their functions and impairing autophagy.

The paper is of interest and suitable for International Journal of Molecular Medicine. The state of art is well described in Introduction. Similarly, the procedures used are in general clearly described. Results and Discussion are clearly written too.

However, I have one main point and a few minor ones to be considered before acceptance:

Main point: in pathologies characterized by aggregating proteins, it has been found that even normal proteins can aggregate when overexpressed (see, for example prions, but also pro-insulin in diabetes). Thus, parallel transfection with a vector encoding normal Htt (Htt-Q23) should have been a better control than the empty vector. Thus, the Authors should explain in Introduction/Discussion why, in their experiments, they decided to use only the empty vector and not the vector encoding the normal Htt, and whether they plan to analyze in the future also the effect of overexpressing the normal protein.

Minor points:

1.      At the beginning of the abstract the mutant Htt is indicated as mtHtt, while then it is always indicated as mHtt;

2.      Results and/or Legends to Figures: even if the activities of some of them have been already reported in the Abstract, please report the main activities of Ned-19, and BAPTA-AM, as well as the meaning of NAADP-AM, also in the legend to Figure 1 and/or in the text that describes the experiments (“Results” Section).

Author Response

In their paper entitled “NAADP-evoked Ca2+ signaling leads to mutant huntingtin aggregation and autophagy impairment in murine astrocytes”, the Authors report that an increase of cytosolic Ca2+ (mediated by the second messenger NAADP, that possibly directly interacts with two pore calcium channels, TPCs) causes mutated huntingtin (mHtt) to aggregate. Interestingly, they also show that mHtt co-localizes with the lysosomes, probably affecting their functions and impairing autophagy.

The paper is of interest and suitable for International Journal of Molecular Medicine. The state of art is well described in Introduction. Similarly, the procedures used are in general clearly described. Results and Discussion are clearly written too.

We thank the reviewer for the comments and suggestions. We appreciated the points indicated about the experiments and the data presented. We have addressed all the questions and we think that the corrections certainly improve the quality of this manuscript.

However, I have one main point and a few minor ones to be considered before acceptance:

Main point: in pathologies characterized by aggregating proteins, it has been found that even normal proteins can aggregate when overexpressed (see, for example prions, but also pro-insulin in diabetes). Thus, parallel transfection with a vector encoding normal Htt (Htt-Q23) should have been a better control than the empty vector. Thus, the Authors should explain in Introduction/Discussion why, in their experiments, they decided to use only the empty vector and not the vector encoding the normal Htt, and whether they plan to analyze in the future also the effect of overexpressing the normal protein.

We thank the reviewer for raising this important question. Reviewing the literature about Huntington´s disease cellular models, many papers demonstrate the effects of overexpression of huntingtin (nonpathogenic). Initially, the overexpression of Htt-Q23 did not affect cell viability when compared to mHtt-Q74 after 48 h of expression, using a PC12 cell model to HD (Karachitos et al., 2016, Front. Oncol., doi: 10.3389/fonc.2016.00238). In another study, the overexpression of GFP-HttQ23 showed to induce the formation of transient oligomers, which does not affect cell viability, while the GFP-mHttQ74 forms larger complexes and increases cell toxicity (Lajoie et al., 2010, Plos One (doi: 10.1371/journal.pone.0015245). Similarly, the overexpression of Htt-Q7 does not form aggregates in mouse striatal cells, while mHtt-Q111 aggregated into large cellular inclusions (Wang et al., 2011, Toxicol. Appl. Parmacol., doi: 10.1016/j.taap.2010.10.032).

Despite many studies describe that huntingtin does not form aggregates and does not affect cell viability, we agreed with the reviewer that cells overexpressing huntingtin could represent a further control group. However the use of the empty vector is also accepted since the cells have the constitutive expression of huntingtin, which may resemble the physiological levels of the protein. For this reason, we performed our experiments using only the GFP-empty vector as control group.

Minor points:

At the beginning of the abstract the mutant Htt is indicated as mtHtt, while then it is always indicated as mHtt;

We corrected the typos as indicated (line 21).

Results and/or Legends to Figures: even if the activities of some of them have been already reported in the Abstract, please report the main activities of Ned-19, and BAPTA-AM, as well as the meaning of NAADP-AM, also in the legend to Figure 1 and/or in the text that describes the experiments (“Results” Section).

We have included in the results section and in the figure 1 legend, the complementary information about the main activities of Ned-19 (antagonist of TPCs) and BAPTA-AM (cytosolic Ca2+ chelator), as well as the meaning of NAADP-AM (nicotinic acid adenine dinucleotide phosphate, acetoxymethyl ester). NAADP-AM is a derivative version of NAADP, more permeant to cell membrane.

Round 2

Reviewer 1 Report

The manuscript is overall improved but I still consider necessary to improve  images of fig.2 demonstrating colocalization, as I suggested in my previous revision.

Author Response

The manuscript is overall improved but I still consider necessary to improve images of fig.2 demonstrating colocalization, as I suggested in my previous revision.

We thank the reviewer for the comments about our manuscript. We replaced and improved the representative images of pEGFP-mHtt Q74 and RFP-TPC2 colocalization presented in Figure 2a.